# Lipopolysaccharide induces mouse translocator protein (18 kDa) expression via the AP-1 complex in the microglial cell line, BV-2

**Shuji Shimoyama**[1,2], **Tomonori Furukawa**[2], **Yoshiki Ogata**[2], **Yoshikazu Nikaido**[2¤a], **Kohei Koga**[2¤b], **Yui Sakamoto**[3], **Shinya Ueno**[1,2], **Kazuhiko Nakamura**[1,3]*

**1** Research Center for Child Mental Development, Hirosaki University Graduate School of Medicine, Hirosaki, Aomori, Japan, **2** Department of Neurophysiology, Hirosaki University Graduate School of Medicine, Hirosaki, Aomori, Japan, **3** Department of Neuropsychiatry, Hirosaki University Graduate School of Medicine, Hirosaki, Aomori, Japan

¤a Current address: Department of Anesthesiology, Hirosaki University Graduate School of Medicine, Hirosaki, Aomori, Japan
¤b Current address: Department of Neurophysiology, Hyogo College of Medicine, Nishinomiya, Hyogo, Japan
* nakakazu@hirosaki-u.ac.jp

**Data Availability Statement:** All relevant data are within the paper and its Supporting Information files.

## Abstract

It has been reported that neuroinflammation occurs in the central nervous system (CNS) in patients with neuropathic pain, Alzheimer's disease and autism spectrum disorder. The 18-kDa translocator protein TSPO is used as an imaging target in positron emission tomography to detect neuroinflammation, and its expression is correlated with microglial activation. However, the mechanism underlying the transcriptional regulation of *Tspo* induced by inflammation is not clear. Here, we revealed that lipopolysaccharide (LPS) -induced *Tspo* expression was activated by the AP-1 complex in a mouse microglial cell line, BV-2. Knockdown of c-Fos and c-Jun, the components of AP-1, reduced LPS-induced *Tspo* expression. Furthermore, the enrichment of Sp1 in the proximal promoter region of *Tspo* was increased in the presence of LPS. In addition, the binding of histone deacetylase 1 (HDAC1) to the enhancer region, which contains the AP-1 site, was decreased by LPS treatment, but there were no significant differences in HDAC1 binding to the proximal promoter region with or without LPS. These results indicated that HDAC1 is involved not in the proximal promoter region but in the enhancer region. Our study revealed that inflammatory signals induce the recruitment of AP-1 to the enhancer region and Sp1 to the proximal promoter region of the *Tspo* gene and that Sp1 may regulate the basal expression of *Tspo*.

## Introduction

Microglia are resident immune cells of the central nervous system (CNS) and play a pivotal role in maintaining the neuronal environment. Resting (ramified) microglia monitor neurons

**Funding:** This work was supported by Hirosaki University Grant for Exploratory Research by Young Scientists and Newly appointed Scientists (to SS), Hirosaki University Institutional Research Grant (to SU and KN) and Grants-in-Aid for Scientific Research (C) #18K08846 (to SU). The funders had no role in study design, data collection and analysis, decision to publish, or preparation of the manuscript.

**Competing interests:** The authors have declared that no competing interests exist.

by "palpating" synapses [1]. Individual microglia have their own territory in the CNS and are distributed throughout the brain so that they do not overlap. In the case of homeostatic disruption in the CNS, microglia transform into an amoeboid (activated) morphology and migrate toward the damaged regions to release neurotoxic or neuroprotective molecules. M1 phenotype microglia produce various types of proinflammatory cytokines, such as IL-1β and TNF-α [2,3], and M2 phenotype microglia also produce neuroprotective and neurotrophic factors, such as BDNF and IGF-1 according to the conditions [4,5]. In addition, if possible, damaged cells are repaired; excessively damaged cells are phagocytosed by activated microglia [6].

Recent studies showed that neuronal inflammation occurs in patients with neuropathic pain, Alzheimer's disease and autism spectrum disorder (ASD) [7–9]. To visualize inflammation in the CNS, positron emission tomography with a radiotracer, such as [$^{11}$C] (*R*) PK-11195 or [$^{11}$C] PBR28, is widely used in animal models or human subjects [10,11]. These radiotracers are specific ligands for the 18-kDa translocator protein TSPO, alternatively called peripheral-type benzodiazepine receptor (PBR) [12]. These tracers are helpful for evaluating the level of neuronal inflammation *in vivo*, thereby determining the treatment plan for CNS disease.

TSPO is expressed in various organs, including glial cells in the CNS, and contributes to the regulation of steroid hormone production by transporting cholesterol, the precursor of pregnenolone, from the cytosol to the mitochondrial intermembrane space [13,14]. Additionally, TSPO is distinct from the central benzodiazepine receptor [γ-aminobutyric acid type A (GABA$_A$) receptor] and plays different roles, for example, in the regulation of cell growth [15] and ATP production [16].

The expression of TSPO is correlated with M1 phenotype microglial activation; however, the mechanism underlying the upregulation of TSPO expression during neuroinflammation has not been elucidated. Previous studies have suggested two possible mechanisms: an increase in the number of microglia in the CNS or of monocytes and macrophages that infiltrate the CNS in response to inflammation [17] and transcriptional activation of *Tspo* in microglia. Expression profiling data (GEO NCBI, #GSE37611) revealed that LPS, a classical inducer of inflammation, increased *Tspo* expression in mouse microglial cell lines [18]; therefore, upregulation of TSPO was the result of transcriptional activation. Accordingly, we focused on transactivation of *Tspo* gene expression in this study, and LPS was used as an inducer of neuroinflammation to elucidate the transcriptional regulation of TSPO.

In the present study, we showed that recruitment of both c-Fos and c-Jun, which are components of the AP-1 complex, to the enhancer region of the *Tspo* gene was increased upon LPS treatment in the microglial cell line BV-2. This evidence was supported by the observation that *Tspo* expression was reduced by knockdown of c-Fos and c-Jun. In addition, we found that the concentration of histone deacetylase 1 (HDAC1) at the AP-1 binding site in the *Tspo* enhancer region was decreased by LPS treatment. These data suggest that LPS-induced *Tspo* gene expression in BV-2 was upregulated by AP-1 activation and that the release of HDAC1 from the AP-1 site was increased by LPS treatment.

## Materials and methods

### Cell culture and chemical treatment

The mouse microglial cell line BV-2 (RRID: CVCL_0182) was purchased from Banca Biologica e Cell Factory (San Martino, Italy). Cells were maintained in DMEM containing 10% FBS and antibiotics and grown at 37°C in an atmosphere containing 5% CO$_2$. Cells were treated with LPS from *E. coli* (O111, Wako, Osaka, Japan) at 1, 10, 100 or 500 ng/ml for various duration. Recombinant murine IFN-γ and IL-1β were purchased from PeproTech (Rocky Hill, NJ, USA).

## Animals and primary microglial cell culture

Male C57BL/6 mice approximately 12 weeks old were used in this study. Mice were housed as described previously [19]. This study was carried out in accordance with the recommendations the guidelines for animal research issued by the Physiological Society of Japan. The experimental procedures were approved by the Animal Research Committee of Hirosaki University (Approval Number: M12007), and all efforts were made to minimize the number of animals used and their suffering. Isolation of microglia from adult mice and primary microglial cell culture were performed as described before [20]. Primary microglial cells were seeded in 35-mm dishes at a density of $1.0 \times 10^4$ cells/dish. Twenty-four hours after seeding, cells were stimulated with 100 ng/ml LPS for 8 hours.

## RNA isolation, cDNA synthesis and quantitative real-time reverse transcription PCR (qRT-PCR)

BV-2 cells were seeded in 35-mm dishes at a density of $1.0 \times 10^5$ cells/dish. After chemical treatment, cells were harvested by centrifugation. Total RNA was extracted from cells using ISO-GEN II (NIPPON GENE, Tokyo, Japan) according to the manufacturer's instructions. cDNA synthesis and qRT-PCR were performed as described previously [19]. The primer sets used in this study are shown in S1 Table. The expression level of each transcript was measured using the standard curve method, *Gapdh* as an internal control.

## Preparation of whole-cell extracts and cell fractionation, Western blotting and densitometric analysis

Whole-cell extracts were prepared as described previously [21]. BV-2 cells were seeded at a density of $1.0 \times 10^5$ cells/35-mm dish (for whole-cell extracts) or at a density of $1.0 \times 10^6$ cells/ 100-mm dish (for cell fractionation). The mitochondrial, membrane, cytosolic and nuclear fractions were prepared according to the following method. After washing the cells with cold PBS (-) containing 1 mM PMSF, cells were harvested via centrifugation. Cells were suspended in hypotonic buffer (10 mM HEPES [pH7.9], 10 mM KCl, 1 mM $MgCl_2$, 0.5 mM DTT and Complete™ Protease Inhibitor Cocktail: Roche, Mannheim, Germany) and incubated on ice for 15 min. Cells were homogenized using a Dounce Tissue Grinder with a tight pestle (WHEATON, Millville, NJ, USA). Nuclei and unhomogenized cells were precipitated via centrifugation at 3,300×g for 5 min at 4°C, and then, the supernatants were then centrifuged at 8,000×g for 15 min at 4°C to obtain the mitochondrial fraction. In addition, the supernatants were centrifuged at 100,000×g for 60 min at 4°C, and the pellets were recovered as the membrane fraction. The resulting supernatants were used as the cytosolic fraction. The nuclear, mitochondrial and membrane fractions were washed 3 times with isotonic buffer (20 mM HEPES [pH7.9], 150 mM KCl, 3 mM $MgCl_2$, 0.5 mM DTT and Complete™ Protease Inhibitor Cocktail) and were then suspended in isotonic buffer containing 1% SDS. The protein concentrations were determined by Bradford protein assay (Bio-Rad, Hercules, CA, USA). Western blotting and densitometric analysis were performed as described previously [19]. The antibodies used in this study are shown in S2 Table.

## Immunocytochemistry

BV-2 cells were seeded on cover glasses at a density of $2.0 \times 10^4$ cells/well (Nunc™ Cell-Culture Treated Multidishes, Thermo Fisher Scientific, Waltham, MA, USA). After chemical treatment, cells were fixed with 4% paraformaldehyde/PBS (-) for 15 min at room temperature and were then permeabilized with 0.1% Triton X-100/PBS (-) for 15 min at room temperature.

After blocking with 1% BSA/PBS (-) for 60 min at room temperature, cells were incubated with the primary antibody at 4˚C overnight. Cells were incubated with the secondary antibody for 2 hours at room temperature, and nuclei were then stained with DAPI for 15 min at room temperature. Cells on the cover glasses were mounted with VECTASHIELD® Mounting Medium (VECTOR LABORATORIES, Burlingame, CA, USA). All steps were carried out in the dark and cells were washed 3 times with PBS (-) or 0.1% Tween 20/PBS (-) after incubation. The antibodies and dilution ratios used in this study are shown in S2 Table. Images were acquired using a confocal laser scanning microscope and analyzed with NIS-Elements AR Analysis software (Nikon, Tokyo, Japan). To label mitochondria, cells were stained with Mito-Tracker Orange® CMTMRos (Thermo Fisher Scientific) in accordance with the manufacturer's instructions.

### Reporter gene assay

The regulatory region of the mouse *Tspo* gene was amplified using forward primers and a common reverse primer with the addition of the Spe I (forward) or Bam HI (reverse) sites as described in S1 Table. The products were ligated to the pMCS-*Gaussia* Luc Vector (Thermo Fisher Scientific), and the sequences were confirmed via sequencing (S1 Table) using a Big-Dye® Terminator v3.1 Cycle Sequencing Kit (Thermo Fisher Scientific); the reaction products were then purified using a Gel Filtration Cartridge (Edge Bio, San Jose, CA, USA). The sample sequences were analyzed using an ABI PRISM® 3100 Genetic Analyzer (Applied Biosystems, Waltham, MA, USA).

Various constructs were transfected into BV-2 cells using Attractene (QIAGEN, Hilden, Germany) in accordance with the manufacturer's instructions. A reporter gene assay was performed using a *Gaussia* Luciferase Glow Assay Kit (Thermo Fischer Scientific) according to the manufacturer's instructions.

### Chromatin immunoprecipitation (ChIP) assay and real-time PCR

A ChIP assay was performed as described previously [21]. The antibodies used in this study are shown in S2 Table. Purified DNA was subjected to real-time PCR with primer pairs specific for the *Tspo* enhancer region containing the AP-1 binding site or the proximal promoter region containing the Sp1 binding site, as described in S1 Table. All quantified values were normalized to that of the corresponding input for each sample using the standard curve method.

### Silencing by RNAi

The c-Fos and c-Jun knockdown experiments were performed using siRNA purchased from Thermo Fisher Scientific (Silencer® Select siRNAs for *Fos*/mouse: #s66198, #s66199; *Jun*/mouse: #s68564; and for Negative Control No.1 [siCont]). siRNAs (10 nM) were transfected into BV-2 cells using Lipofectamine RNAiMAX (Invitrogen, Carlsbad, CA, USA) in accordance with the manufacturer's instructions.

### Statistical analyses

All numerical data are presented as the means ± S. Ds, with n = 3 (qRT-PCR, Western blotting and ChIP assay) or n = 4 (reporter gene assay). Comparisons were assessed using Student's t-test or ANOVA. A value of $p < 0.05$ was considered statistically significant. Statistical analyses were performed using StatPlus:mac Pro software (AnalystSoft, Walnut, CA, USA).

## Results

### LPS treatment induced *Tspo* expression in BV-2 cells

To determine the optimum assay conditions, BV-2 cells were treated with LPS for various durations and at various concentrations. The expression of *Tspo* mRNA was induced by a high concentration of LPS (100 or 500 ng/ml), peaked at 8 hours and gradually returned to normal. By contrast, the expression of *Tspo* mRNA did not increase at low concentrations of LPS (1 or 10 ng/ml). Under the same conditions, the expression of pro-inflammatory mediators such as *Tnf-α*, *Il-1β* and *Il-6* was also upregulated by LPS stimulation (Fig 1A). Furthermore, the transcriptional activation of *Tspo* was induced by LPS stimulation in primary microglial cells from adult mice (Fig 1B).

Subsequent to analysis of the *Tspo* mRNA expression levels, Western blotting and densitometric analysis were also performed to the measure protein levels (Fig 1C and 1D). The protein expression of TSPO was correlated with its mRNA expression. In addition, the results of Western blotting and immunocytochemistry revealed that TSPO expression was increased at high concentrations of LPS (Fig 1E and S1 Fig). These results indicated that LPS was a potent inducer of *Tspo* in mouse microglial cells, as reported previously [22]. In addition, we examined whether proinflammatory mediators such as IFN-γ or IL-1β induce the expression of *Tspo*. BV-2 cells were stimulated with 100 ng/ml IFN-γ and/or IL-1β for 8 hours. The expression of *Tspo* mRNA was also induced by administration of IFN-γ, IL-1β, or both (S2 Fig).

### LPS treatment increased TSPO expression in the mitochondrial and membrane fractions

The TSPO protein is mainly localized in mitochondria in the adrenal gland homogenate [23]. To investigate whether TSPO in microglial cells is localized in mitochondria and whether the intracellular localization of TSPO is affected by LPS treatment, subcellular fractionation analysis was performed. BV-2 cells were collected, and cell fractions were prepared for Western blotting after stimulation with 100 ng/ml LPS for 8 hours. First, the purity of the fractions was confirmed by fraction markers, such as COXIV, the alpha1 subunit of the Na$^+$/K$^+$ ATPase, GAPDH and Lamin-B1, and there was no contamination or variation in expression between the control and LPS-treated groups in the same fractions. Similar to the fractionation analysis results, TSPO expression was significantly increased in the mitochondrial fraction by LPS treatment (Fig 2A and 2B, *p* = 0.03). Interestingly, TSPO was also localized in the membrane fraction in the control group and was 1.94-fold higher in the LPS-treated group than in the control group (*p* = 0.005). To verify whether TSPO is localized in the plasma membrane, TSPO, the Na$^+$/K$^+$ ATPase and mitochondria were stained by immunocytochemistry and analyzed by laser confocal microscopy. As previously reported, almost all TSPO protein was localized in the mitochondrial outer membrane. However, some fluorescence signals corresponding to TSPO protein were detected in non-mitochondrial regions in both the absence and presence of LPS (Fig 2C, upper, arrowhead). In addition, some TSPO signals were colocalized with or very close to Na$^+$/K$^+$ ATPase signals (Fig 2C, lower, arrowhead). These results indicated that a small amount of TSPO was localized in the plasma membrane.

### The transcription factor AP-1 is the candidate transcriptional activator of *Tspo* after LPS treatment

Prior to the construction of vectors for the reporter gene assay, the *Tspo* regulatory sequence was analyzed using ChIP-seq data in the UCSC Genome Browser (http://genome.ucsc.edu/ENCODE/) and prediction tools for transcription binding sites, such as STAMP (http://www.

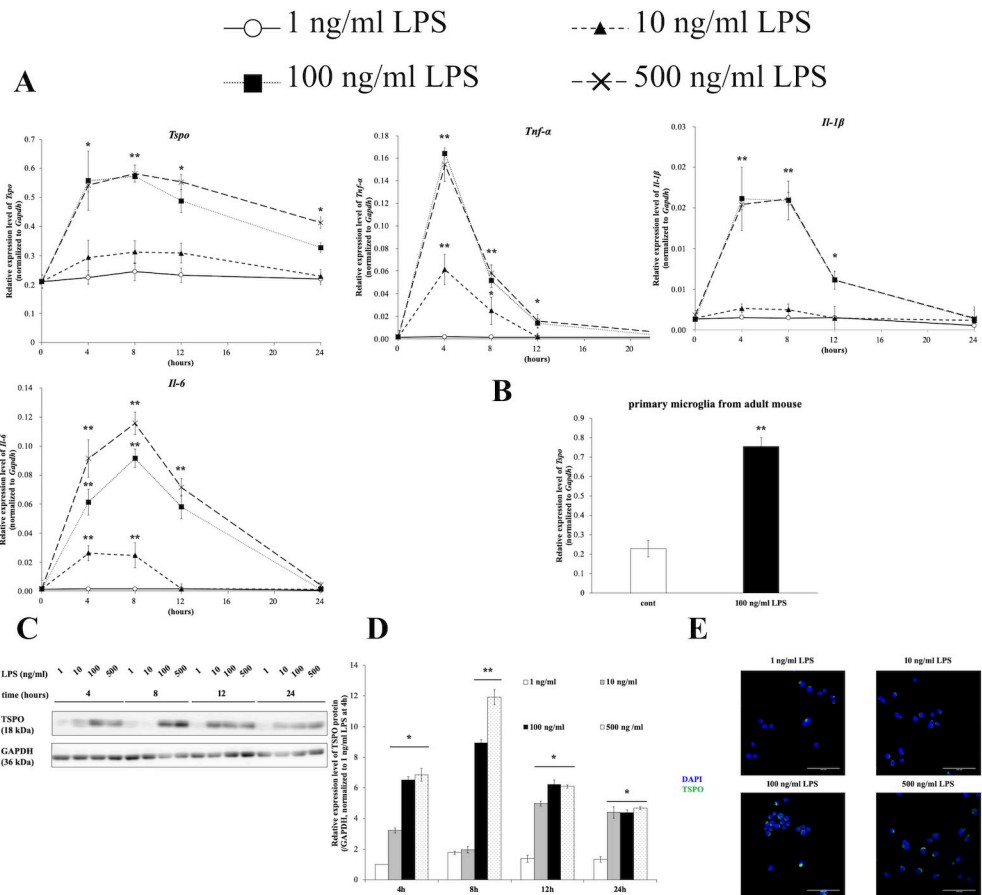

**Fig 1. LPS induced *Tspo* expression in the microglial cell line BV-2. (A)** BV-2 cells were treated for the indicated time and with the indicated dose of LPS, and RNA was then extracted and subjected to qRT-PCR for *Tspo*, *Tnf-α*, *IL-1β*, *IL-6* and *Gapdh*. The expression level of each mRNA was measured and normalized to that of *Gapdh* mRNA. The asterisks indicate statistically significant differences compared to 1 ng/ml LPS (n = 3, $^*p<0.05$, $^{**}p<0.01$). **(B)** Primary microglial cells from adult mice were stimulated with 100 ng/ml LPS for 8 hours and were then subjected to qRT-PCR. **(C)** BV-2 cells were treated as described above. The protein expression level of TSPO was measured using Western blot analysis. **(D)** Densitometric analysis of TSPO expression using ImageJ. The expression value was measured and normalized to that of GAPDH as the internal control. The asterisks indicate statistically significant differences (n = 3, $^*p<0.05$, $^{**}p<0.01$). **(E)** Microscopy images showing TSPO expression. BV-2 cells were stimulated with 1, 10, 100 or 500 ng/ml LPS for 8 hours and immunocytochemistry was then carried out. Images of nuclear (DAPI, blue) and TSPO (green) staining were acquired at 40× magnification. Scale bar: 100 μm.

benoslab.pitt.edu/stamp/), JASPAR (http://jaspardev.genereg.net/) and DBCLS Galaxy (http://galaxy.dbcls.jp/). The presumed transcription factors and their binding sites were identified via a comprehensive assessment using several tools and are shown in Fig 3A. To determine which transcription factor and *cis* element regulates *Tspo* expression during LPS treatment, serial deletion constructs were designed for the *Tspo* reporter gene assay according to the schematic in Fig 3A. Forty-eight hours after transfection of each vector, BV-2 cells were treated with vehicle or 100 ng/ml LPS for an additional 8 hours, and a reporter gene assay was then performed. As shown in Fig 3B, deletion of the AP-1 site (-517) significantly reduced luciferase activity; however, basal expression was not altered. These results indicated that AP-1 was the candidate transcriptional activator of *Tspo* during LPS stimulation.

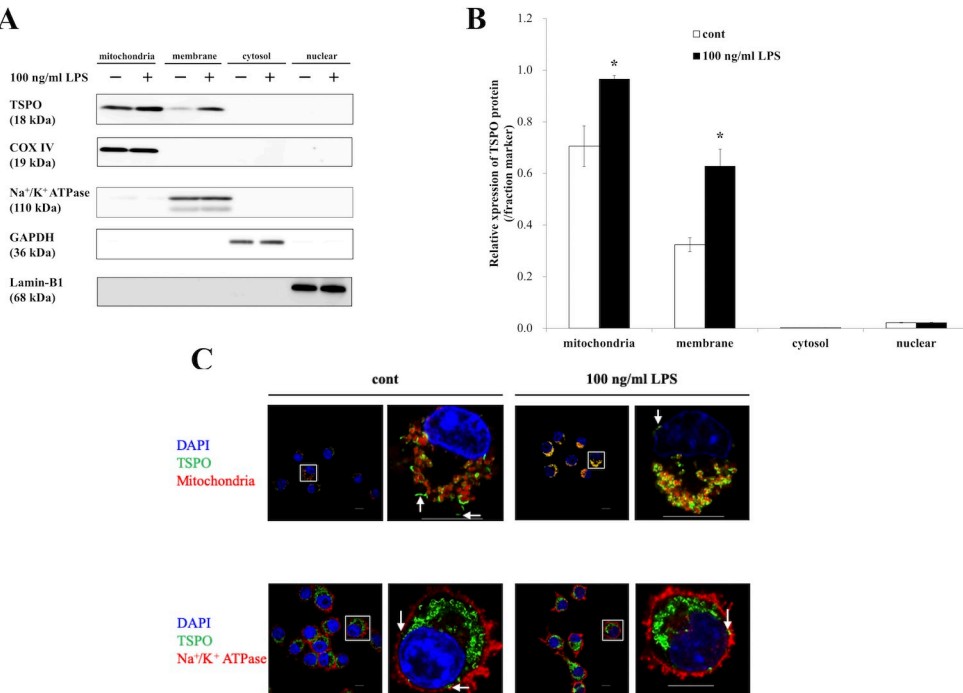

**Fig 2. TSPO protein expression was increased in the mitochondrial and membrane fractions by LPS treatment.**
**(A)** Western blot analysis of the TSPO protein and fraction markers with or without 100 ng/ml LPS treatment for 8 hours. COXIV, ATPase, GAPDH and Lamin-B1 were used as the internal markers of mitochondria, membranes, the cytosol and nuclei, respectively. **(B)** The relative expression of the TSPO protein in each fraction was measured and normalized using each fraction marker. The asterisks indicate statistically significant differences from the control in the same fraction (n = 3, $p < 0.05$). **(C)** Images showing TSPO localization in BV-2 cells. BV-2 cells were stimulated with 100 ng/ml LPS for 8 hours, and immunocytochemistry was then carried out. Images of mitochondrial (red, upper), Na$^+$/K$^+$ ATPase (red, lower), TSPO (green) and nuclear (DAPI, blue) staining were acquired at 100× magnification with a confocal microscope. Arrowhead: TSPO protein signals in non-mitochondrial regions. Scale bar: 10 µm.

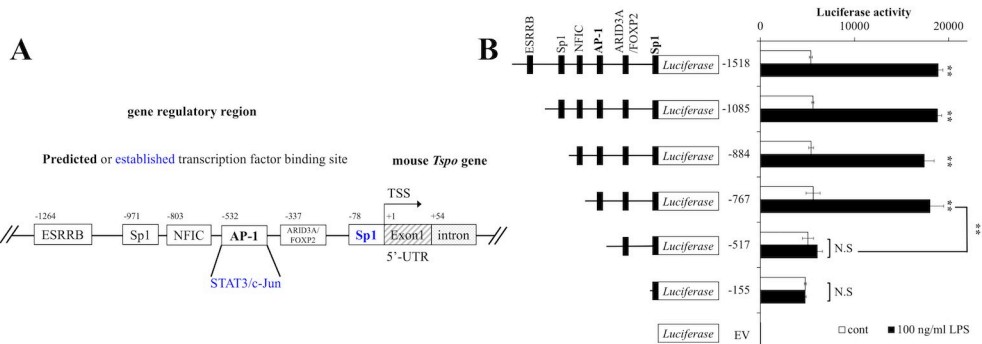

**Fig 3. Involvement of AP-1 in the regulation of *Tspo* expression by LPS stimulation. (A)** Schematic of the predicted and established transcription factor binding sites upstream of the mouse *Tspo* gene. The blue letters indicate established transcription factors that regulate the mouse *Tspo* genes based on a previous study. TSS: Transcriptional start site, UTR: Untranslated region **(B)** Forty-eight hours after transfection of the indicated vectors, BV-2 cells were treated with 100 ng/ml LPS for an additional 8 hours. Luciferase activity was measured as described in the Materials and methods section. The asterisks indicate statistically significant differences (n = 4, $^{**}p < 0.01$).

## AP-1, Sp1 and HDAC1 regulated *Tspo* expression

Based on the results of the reporter gene assay and previous studies [24,25], we examined via a ChIP assay whether the enrichment of AP-1 at the binding site in the *Tspo* regulatory region was promoted by LPS treatment. A previous study showed that *TSPO* expression is regulated by the binding of Sp1, Sp3 and/or Sp4 to the proximal promoter region in human breast cancer cell lines, and a similar sequence was observed in the mouse *Tspo* gene [26]. In addition, there is the possibility that HDAC1 acts as an inhibitory factor of *Tspo* expression by binding to Sp1 in a transcriptionally repressed state [21,27]. Furthermore, since few studies have examined the interaction between HDAC1 and Sp3 or Sp4, we excluded Sp3 and Sp4 from this study and focused on the interaction between Sp1 and HDAC1 in the proximal promoter region of the *Tspo* gene. Based on these observations, the primer sets used for the ChIP assay were designed as shown in the schematic in Fig 4A. The ChIP experiment revealed that the enrichment of the AP-1 components c-Fos and c-Jun at the AP-1 binding site was increased by approximately 3.83-and 2.88-fold, respectively, in the presence of LPS (Fig 4B). In the proximal promoter region, the enrichment of RNA polymerase II (RNAP II) and Sp1 was increased at the time of transcriptional activation compared to that in control cells (Fig 4C). These results clearly showed that the binding of AP-1 and Sp1 the upstream region of the *Tspo* gene was increased by LPS treatment. Interestingly, after LPS treatment, enrichment of HDAC1 was decreased in the enhancer region, which contains the AP-1 site, but enrichment of HDAC1 was not altered in the proximal promoter region.

## Silencing of c-Fos and c-Jun decreased *Tspo* expression

To confirm the involvement of AP-1 in Tspo expression, silencing of c-Fos and c-Jun was performed via RNAi. First, Western blot analyses were performed to evaluate the knockdown efficiency of c-Fos and c-Jun. Forty-eight hours after transfection of c-Fos siRNA, c-Jun siRNA or

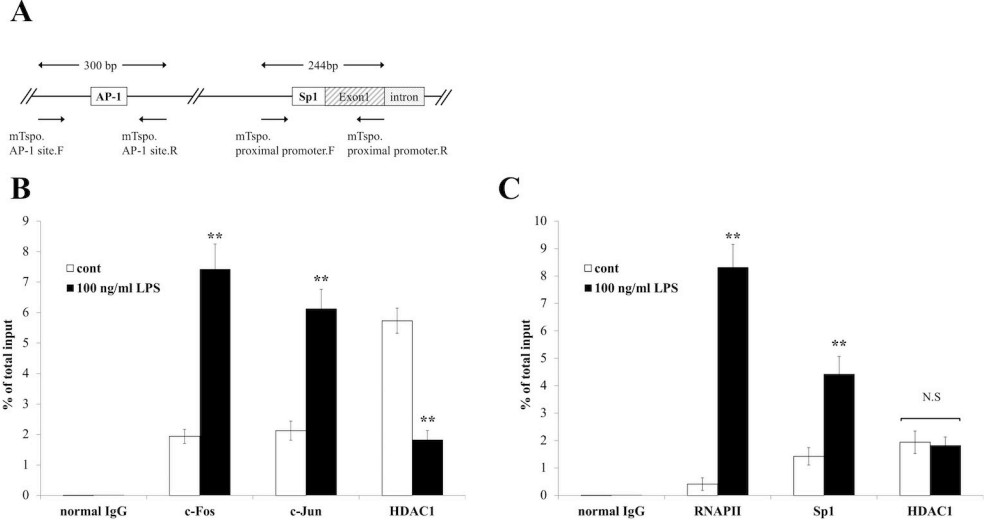

**Fig 4. Enrichment of AP-1 and Sp1 was increased and enrichment of HDAC1 was decreased in the *Tspo* gene regulatory region by LPS treatment. (A)** Schematic of the upstream region of the mouse *Tspo* gene and the positions of the primers used for the ChIP assay. **(B)** Enrichment of c-Fos, c-Jun and HDAC1 at the AP-1 binding site. BV-2 cells were treated with 100 ng/ml LPS for 2 hours. Then, ChIP analyses were performed as described in the Materials and methods section. **(C)** Enrichment of RNAP II, Sp1 and HDAC1 in the proximal promoter region. The Y-axis indicates the quantity of PCR products normalized to that of the input for each sample. The asterisks indicate statistically significant differences from the control (n = 3, $p < 0.01$).

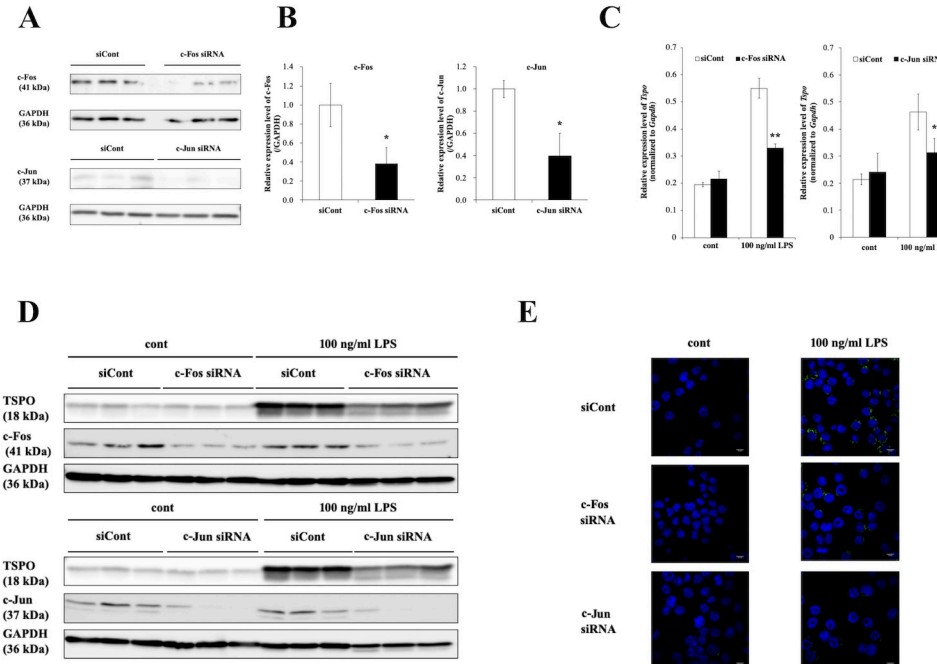

**Fig 5. Knockdown of c-Fos and c-Jun reduced *Tspo* expression. (A)** Efficiency of c-Fos and c-Jun knockdown by siRNAs. Forty-eight hours after transfection of siRNAs against c-Fos or c-Jun or the control siRNA, BV-2 cells were harvested and used to measure the expression of c-Fos and c-Jun by Western blotting. **(B)** Densitometric analysis of the bands corresponding to the c-Fos (left) and c-Jun (right) proteins. The asterisks indicate statistically significant differences compared with siCont (n = 3, $p<0.05$). **(C)** Relative expression of *Tspo* mRNA with or without 100 ng/ml LPS treatment and siRNA against c-Fos or c-Jun. Forty-eight hours after transfection of c-Fos siRNA, c-Jun siRNA or siCont, BV-2 cells were treated with 100 ng/ml LPS for an additional 8 hours, and total RNA was then isolated and subjected to qRT-PCR. The asterisks indicate statistically significant differences compared with the control (n = 3, $^*p<0.05$, $^{**}p<0.01$). **(D)** BV-2 cells were treated as described above. The protein expression level of TSPO was measured via Western blot analysis. **(E)** Microscopy images showing TSPO expression. Images of nuclear (DAPI, blue) and TSPO (green) staining were acquired at 100× magnification with a confocal microscope. Scale bar: 10 μm.

siCont, cells were harvested and subjected to Western blot analysis of c-Fos and c-Jun. As shown in Fig 5A and 5B, the expression of c-Fos and c-Jun was clearly decreased by siRNA compared to siCont. Under these conditions, cells were stimulated with 100 ng/ml LPS for an additional 8 hours and analyzed to measure the expression level of *Tspo* mRNA. The results of qRT-PCR analysis revealed that knockdown of c-Fos or c-Jun repressed the LPS-promoted expression of *Tspo* (Fig 5C). Similar results were obtained at the protein level for TSPO (Fig 5D and 5E). Therefore, c-Fos and c-Jun are essential transcriptional activators of LPS-induced *Tspo* regulation in mice.

## Discussion

In this study, LPS-induced gene expression of *Tspo* in a microglial cell line was examined. Our experiments clearly demonstrated that AP-1, Sp1 and HDAC1 are involved in the transactivation of *Tspo* expression. Specifically, the release of HDAC1 from the enhancer region of the *Tspo* gene in the presence of LPS was a novel finding.

The transcription factors c-Jun and STAT3 bind to the enhancer region of the *Tspo* gene in the mouse cell lines MA-10 and NIH-3T3 via PKCε activation through phorbol 12-myristate 13-acetate (PMA) stimulation [25]. PKCε activation via PMA activates the Raf1, MEK1/2 and ERK1/2 pathways. This signal transduction promotes *Tspo* gene expression following the

activation and DNA binding of STAT3 and c-Jun. Our results indicating that LPS stimulation led to the binding of c-Fos and c-Jun to the enhancer region were partially consistent with those of previous studies. In general, LPS binds directly to Toll-like receptor 4 (TLR4) [28] and activates downstream factors such as mitogen-activated protein kinases (MAPKs) [29,30]. The MKK3/6 and p38 MAPK pathways induce phosphorylation of c-Fos and enhance transcriptional activity [31]. Additionally, c-Jun phosphorylation is induced by TLR4 signaling via the MKK4/7 JNK pathways, and dimerization of c-Jun and c-Fos is promoted [32,33]. Collectively, these results indicate that LPS treatment is likely to activate the AP-1 complex in a manner mediated by several transduction pathways, such as the MAPK pathway, and that *Tspo* expression is consequently promoted by the binding of the AP-1 complex to DNA. In contrast to the facilitation of AP-1 binding, the enrichment of HDAC1 at the AP-1 binding site was decreased by LPS treatment. The reduction in HDAC1 enrichment in the enhancer region by LPS treatment possibly led to hyperinduction of *Tspo* expression via recruitment of AP-1 or another transcriptional activator. A recent study revealed that deletion of sequences between -593 and -520 bp from the TSS, which contains the AP-1, Stat3 and Ets.2 *cis* elements, abolished the transactivation of *Tspo* gene expression in BV-2 and Raw-264.7 macrophages [34]. Since our results were consistent with those of that recent report, it is convincingly concluded that the AP-1 and Sp1 are involved in the LPS-induced transcriptional regulation of *Tspo*. In addition to the involvement of AP-1 and Sp1, Rashid *et al* reported that the Pu.1 and its binding motif, Ets, play an important role in the transactivation of *Tspo* gene expression. Prediction tools for transcription binding sites were used in this study; however, Pu.1/Ets was not identified as a candidate transcription factor for the regulation of the *Tspo* gene. Contrary to the prediction results, the Ets sequence (5'-GGAA-3') is located 3 bp from the AP-1 binding site and acts as a transcriptional activator of *Tspo* in response to LPS stimulation.

In addition to LPS stimulation, treatment with IFN-γ and IL-1β also induced *Tspo* transcription. Generally, IFN-γ activates the JAK-STAT pathway, and STAT acts as a transcriptional activator of inflammatory genes. The consensus sequence of STAT3 is located the upstream region of the *Tspo* gene, which overlaps with the AP-1 and Ets sites [25]. The IL-1β signaling pathway activates several transcription factors, including AP-1, via MAPKs such as p38, JNK and ERK1/2 [35]. These lines of evidences indicate that the general inflammatory response induces the expression of *Tspo* in microglial cells via the activation of transcription factors such as AP-1 or STATs.

Sp1, which belongs to the $C_2H_2$-type zinc-finger protein family, binds to GC-rich motifs and regulates numerous genes [36]. A previous study showed that there are several GC-rich sequences in the proximal promoter region of the mouse *Tspo* gene [26]. In this study, LPS treatment of BV-2 cells increased the enrichment of Sp1 in the proximal promoter region of *Tspo*. This evidence suggests that Sp1 binds to GC-rich sequences to regulate various genes by interacting with a transcriptional activator or repressor such as CBP/p300 or HDAC1 [37,38]. In the proximal promoter region of the *Tspo* gene, Sp1 enrichment was increased by LPS treatment, but HDAC1 enrichment was not affected. This result indicated that HDAC1 localized in the proximal promoter region was not involved in the pathway downstream of TLR4 activation by LPS. Furthermore, the finding that *Tspo* was expressed to some extent even under the control condition suggested that HDAC1 might minimally bind to the proximal promoter region under physiological conditions.

The expression of TSPO can be upregulated in noninflammatory conditions. According to the Human Brain Transcriptome database (http://hbatlas.org/) [39], human *TSPO* expression increases during the embryonic to the prenatal period and decreases during the neonatal period, and the *TSPO* expression level in the neonatal period is maintained throughout life. These data suggest that the functions of TSPO, such as increasing cholesterol transport from

the cytosol to mitochondria and promoting pregnenolone production, might be important for fetal development. In addition, there is a possibility that the increase in *TSPO* expression during the neural developmental stage might be reflective of activated microglia. During neural development, microglia are activated and maintain CNS homeostasis by removing apoptotic cells, myelin debris and other endogenous redundant debris [40]. It is unknown which transcription factor or factors regulate the transcription of the *TSPO* gene in the fetal period; however, the unknown pathway might activate AP-1 and Sp1 or a different transcription factor.

In this study, we found that TSPO was not only localized in mitochondria but also in the plasma membrane fraction, as evidenced by the cell fractionation, Western blotting and immunocytochemical analysis results. The results of Western blot analysis indicated that not only the mitochondrial fraction but also in the membrane fraction contained TSPO. Consistent with this finding, the immunocytochemical results revealed that TSPO was mainly colocalized with mitochondria and that small amounts of TSPO signals were not distant from but were closely localized with $Na^+/K^+$ ATPase signals regardless of LPS stimulation. Generally, the $Na^+/K^+$ ATPase localizes to the transmembrane region and is used as a marker of the plasma membrane. However, the plasma membrane is not solely constituted by the $Na^+/K^+$ ATPase. Consequently, the TSPO signals were not completely colocalized with the plasma membrane signals. In addition to mitochondria and the plasma membrane, other organelles appeared to feature TSPO immunostaining signals. Thus, as it is reported that not all of the TSPO protein colocalizes with mitochondria, it is possible that TSPO localizes in organelles other than mitochondria [22]. Further research is needed to assess the localization and physiological function of TSPO induced by LPS stimulation.

In patients with ASD, the binding potential of [$^{11}$C] (*R*) PK-11195 to TSPO is increased in multiple brain regions, such as the cerebellum, brainstem and anterior cingulate cortex (ACC) [9]. These results indicate that the activation of microglia, i.e., neuroinflammation, and the expression of TSPO are increased. However, the conditions in our *ex vivo* study using a microglial cell line and LPS differed from the those of the CNS in patients with ASD; therefore, further investigation is needed to clarify the transcriptional regulation of *Tspo* using an animal model of ASD. It is necessary to elucidate whether this alteration in TSPO is due to transcriptional activation via AP-1 or a different transcriptional factor or factors in patients with ASD or animal model of ASD.

It is also important to clarify whether expressed TSPO leads to biological protection against or promotion of inflammation in the CNS. Because TSPO expression is increased under inflammatory conditions, it is thought that TSPO may act as a mediator of inflammation [41]. However, a recent study reported that TSPO acts as a negative regulator of inflammation [22]. Furthermore, TSPO overexpression rescues LPS-induced cognitive impairment. TSPO reduced inflammatory cytokines production and microglial activation, and neurogenesis in the hippocampus was promoted. Consequently, LPS-induced cognitive dysfunction was ameliorated [42]. TSPO may play a neuro-protective role in inflammation, and thus, it could be expected that elucidating the mechanism of TSPO induction under conditions other than inflammation could lead to the development of molecular targeted drugs.

## Supporting information

**S1 Table. Oligonucleotide sequences used in this study.** Oligonucleotide sequences used for real-time PCR and vector construction; sequencing primers used for the vectors and ChIP assay.
(XLSX)

**S2 Table. List of antibodies used in this study.** Antibodies used in this study are shown with their supplier, product number and experimental procedure. WB: Western blotting, ICC: Immunocytochemistry, ChIP: Chromatin immunoprecipitation
(XLSX)

**S1 Fig. Microscopy images showing TSPO expression.** BV-2 cells were stimulated with 1, 10, 100 or 500 ng/ml LPS for 4, 8, 12 or 24 hours and immunocytochemistry was then carried out. Images of nuclear (DAPI, blue) and TSPO (green) staining were acquired at 40× magnification. Scale bar: 100 μm.
(TIFF)

**S2 Fig. IFN-γ and IL-1β induced *Tspo* expression in the microglial cell line, BV-2.** BV-2 cells were treated with 100 ng/ml IFN-γ and/or IL-1β for 8 hours and RNA was then extracted and subjected to qRT-PCR. The mRNA levels were measured and normalized to those of *Gapdh* mRNA. The asterisks indicate statistically significant differences compared to the control (n = 3, $^*p<0.05$, $^{**}p<0.01$).
(TIFF)

## Acknowledgments

We are thankful to Sachiko Kamikawa and Noriaki Kawai for their technical assistance. We would like to thank Springer Nature Author Services (https://authorservices.springernature.com/) for English language editing. This work was performed in part at the Institute for Animal Experimentation, Hirosaki University.

## Author Contributions

**Conceptualization:** Shuji Shimoyama, Shinya Ueno, Kazuhiko Nakamura.

**Data curation:** Shuji Shimoyama.

**Formal analysis:** Shuji Shimoyama, Tomonori Furukawa.

**Funding acquisition:** Shuji Shimoyama, Shinya Ueno, Kazuhiko Nakamura.

**Investigation:** Shuji Shimoyama.

**Methodology:** Shuji Shimoyama.

**Project administration:** Shinya Ueno, Kazuhiko Nakamura.

**Supervision:** Yoshiki Ogata, Yoshikazu Nikaido, Kohei Koga, Yui Sakamoto, Shinya Ueno, Kazuhiko Nakamura.

**Validation:** Shuji Shimoyama, Tomonori Furukawa.

**Visualization:** Shuji Shimoyama, Tomonori Furukawa.

**Writing – original draft:** Shuji Shimoyama, Tomonori Furukawa, Yoshiki Ogata, Yoshikazu Nikaido, Kohei Koga, Yui Sakamoto, Shinya Ueno, Kazuhiko Nakamura.

**Writing – review & editing:** Shuji Shimoyama, Tomonori Furukawa, Yoshiki Ogata, Yoshikazu Nikaido, Kohei Koga, Yui Sakamoto, Shinya Ueno, Kazuhiko Nakamura.

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
