## [Decision Letter · Decision Letter 0]

17 Jul 2019

PONE-D-19-15932

AP-1 complex induces mouse translocator protein (18 kDa) expression by lipopolysaccharide in microglial cell line, BV-2

PLOS ONE

Dear Professor Nakamura,

Thank you for submitting your manuscript to PLOS ONE. After careful consideration, we feel that it has merit but does not fully meet PLOS ONE’s publication criteria as it currently stands. Therefore, we invite you to submit a revised version of the manuscript that addresses the points raised during the review process.

We would appreciate receiving your revised manuscript by Aug 31 2019 11:59PM. To enhance the reproducibility of your results, we recommend that if applicable you deposit your laboratory protocols in protocols.io, where a protocol can be assigned its own identifier (DOI) such that it can be cited independently in the future. For instructions see: http://journals.plos.org/plosone/s/submission-guidelines#loc-laboratory-protocols

We look forward to receiving your revised manuscript.

Kind regards,

Leo T.O. Lee, Ph.D.

Academic Editor

PLOS ONE

Journal Requirements:

Reviewers' comments:

Reviewer's Responses to Questions

**Comments to the Author**

1. Is the manuscript technically sound, and do the data support the conclusions?

Reviewer #1: Yes

Reviewer #2: Yes

Reviewer #3: Partly

2. Has the statistical analysis been performed appropriately and rigorously? 

Reviewer #1: Yes

Reviewer #2: Yes

Reviewer #3: N/A

3. Have the authors made all data underlying the findings in their manuscript fully available?

Reviewer #1: Yes

Reviewer #2: Yes

Reviewer #3: Yes

4. Is the manuscript presented in an intelligible fashion and written in standard English?

Reviewer #1: Yes

Reviewer #2: Yes

Reviewer #3: No

5. Review Comments to the Author

Reviewer #1: The manuscript could be improved by considering the following points

1. I think it would be important to include something about PU.1 in the discussion (The binding motif of PU.1 is in enhancer region (-593-520) and lies 3 bp apart from the AP1 core binding motif). The expression pattern of TSPO in microglia is very specific, and therefore Lineage dependent transcription factors (LDTFs) such as PU.1 have to play a role (Especially since we have already provided evidence of strong Pu.1 enrichment in this region). Infact, the initial steps of enhancer selection in closed chromatin regions containing regularly positioned nucleosomes involves the binding of LDTFs. This binding results in the depletion/sliding of nucleosomes to expose enhancer DNA sequences. Indeed, in microglia/macrophages, enhancers controlling endotoxin-stimulated gene expression are almost invariably bound by the lineage dependent Ets transcription factor Pu.1 (Ghisletti et al., 2010; Holtman et al., 2017). These LDTFs bound to enhancers induce histone modifications (such as H3K4me1) associated with a primed state of activity (Smale and Natoli, 2014). Histone mark H3K4me1 is a core chromatin signature of primed enhancers and serves as a beacon for signal-dependent effectors of signaling pathways such as nuclear factor-κB (NFκB), interferon responsive factors (IRFs) and activator protein 1 (AP-1) (Ghisletti et al., 2010; Heinz et al., 2010; Smale and Natoli, 2014).

Ghisletti, S., Barozzi, I., Mietton, F., Polletti, S., De Santa, F., Venturini, E., Gregory, L., Lonie, L., Chew, A., Wei, C.-L., et al. (2010). Identification and Characterization of Enhancers Controlling the Inflammatory Gene Expression Program in Macrophages. Immunity 32, 317–328.

Heinz, S., Benner, C., Spann, N., Bertolino, E., Lin, Y.C., Laslo, P., Cheng, J.X., Murre, C., Singh, H., and Glass, C.K. (2010). Simple combinations of lineage-determining transcription factors prime cis-regulatory elements required for macrophage and B cell identities. Mol. Cell 38, 576–589.

Holtman, I.R., Skola, D., and Glass, C.K. (2017). Transcriptional control of microglia phenotypes in health and disease. J. Clin. Invest. 127, 3220–3229.

Smale, S.T., and Natoli, G. (2014). Transcriptional control of inflammatory responses. Cold Spring Harb. Perspect. Biol. 6, a016261.

2. It would be important to comment on differences observed in TSPO expression following knockdown of cFos and cJun. In a recent paper (Raashid et al.Biochim Biophys Acta Gene Regul Mech. 2018 Dec;1861(12):1119-1133.), decreased TSPO expression was only observed following a combined knockdown of cJun and cFos. In the current paper, the authors report a decrease in TSPO expression when either cJun or cFos is knockdown. Is this due to cell-line differences? (I noticed they used a different BV-2 cell-line which was maintained in DMEM medium).

3. In paragraph 4 of the discussion, Sp1 transcription factor is discussed. There is no mention of either Sp3 or Sp4, despite Rashid et al. showing that these factors bind the GC-boxes in the proximal promoter (Sp1, Sp3 and Sp4 share common binding site motifs) and that knockdown of either Sp3 or Sp4 significantly reduces TSPO promoter activity in BV-2 microglia.

4. In paragraph 5 of the discussion, the authors state “ human TSPO expression increases during the embryo to prenatal period and decreases during the neonatal period, and the TSPO expression level in the neonatal period is maintained throughout life. These data suggest that the TSPO functions, such as increasing cholesterol transport from cytosol to mitochondria and promoting pregnenolone production, might be important for fetal development.” While this statement might be true, I think it also important to acknowledge that the increase in TSPO expression during development might simply be a reflection of an activated microglia state. During development, microglia are predominantly in the amoeboid state as they play a major role in removing excessive produced apoptotic neurons and large amounts of extracellular debris from the developing CNS.

Reviewer #2: The authors adequately addressed the issues from the original review. The paper describes a data-supported mechanism for transcriptional regulation of TSPO in LPS-treated microglia. A few minor points:

1) Regarding M1/M2 classification - it should be noted that the range of microglial activation phenotypes is complex and that the M1/M2 classification in the paper is being used for the sake of simplicity.

2) The antibodies and oligonucleotide sequences used can be placed in supplemental information.

3) There were a few grammatical errors throughout the manuscript.

Reviewer #3: In this manuscript the authors described a potential mechanism of the increased TSPO expression after LPS treatment which were regulated by AP-1 complex and HDAC1 in BV-2 microglia cell at the transcriptional level. I have some specific issues that need clarification.

1. The title should be replaced as “Lipopolysaccharide induces mouse translocator protein (18 kDa) expression by AP-1 complex in microglial cell line, BV-2”.

2. As the authors stated in the manuscript, the concentration of HDAC1 (a transcriptional repressor) was decreased in the enhancer region of TSPO at the AP-1 binding site after LPS treatment. I have concerns about the potential effects of HDAC1 intervention on TSPO expression or LPS induced inflammatory reaction.

3. Writing of the paper might be improved to meet sufficient quality.

6. PLOS authors have the option to publish the peer review history of their article (what does this mean?). If published, this will include your full peer review and any attached files.

Reviewer #1: No

Reviewer #2: No

Reviewer #3: No

---

## [Author Response · Author response to Decision Letter 0]

18 Aug 2019

Reviewer 1

1. Thank you for the constructive comments to improve our study. We should have mentioned about Pu.1/Ets.2 in the discussion. We added the sentences as below (P17, line 15-21).

 “... transcriptional regulation of Tspo. In addition to the involvement of AP-1 and Sp1, Rashid et al reported that the Pu.1 and its binding motif, Ets, play an important role in the transactivation of Tspo gene expression. Prediction tools for transcription binding sites were used in this study; however, Pu.1/Ets was not identified as a candidate transcription factor for the regulation of the Tspo gene. Contrary to the prediction results, the Ets sequence (5’-GGAA-3’) is located 3 bp from the AP-1 binding site and acts as a transcriptional activator of Tspo in response to LPS stimulation.”

2. Thank you for the constructive comments to improve our study. We also wondered the differences between our results and a recent study (Rashid et al, Biochim Biophys Acta. Gene Regul Mech, 2018). Batarseh et al reported that PMA-induced Tspo expression was reduced by the solely knockdown of c-Jun in MA-10 cells and NIH-3T3 cells (ref#.25, Figure 4 and 6), hence, it is thought to be that the expression of Tspo was inhibited by the knockdown of either of c-Fos or c-Jun. In case of our study, the protocols such as cell and its culture condition, the dose or time of LPS stimulation and the measurement of gene expression by qRT-PCR or Western blotting were diffed from previous studies, the results were not completely consistent with those studies.

3. Thank you for your comments. Rashid et al (Ref. #34) and Batarseh et al (Ref. #26) reported that the enrichment of Sp1, Sp3 and Sp4 were increased at the GC-boxes in the proximal and distal promoter region of Tspo gene and the knockdown experiments of Sp3 or Sp4 significantly reduced the transactivation of Tspo, hence, it is considered to be a certain phenomenon that the involvement of Sp family in the Tspo gene expression.

Previous studies showed that Sp1 and HDAC1 forms a transcriptional repressor complex at the transcriptional inactive state. Upon the stimulation, the enrichment of Sp1 did not affect, however, HDAC1 was released from Sp1 and replaced to CBP/p300 (Ref. #21 and 27).

We hypothesized that the same molecular mechanism was occurred in the transcriptional regulation of Tspo gene. Furthermore, since there were few studies that the interaction between HDAC1 and Sp3 or Sp4, we excluded Sp3 and Sp4 from this study. The additional information is shown in the results section as described below. (P14, line1-4)

“... a, transcriptionally repressed state [21,27]. Furthermore, since few studies have examined the interaction between HDAC1 and Sp3 or Sp4, we excluded Sp3 and Sp4 from this study and focused on the interaction between Sp1 and HDAC1 in the proximal promoter region of the Tspo gene.”

4. Thank you for the constructive comments to improve our study. As you pointed out, we should have mentioned about the activation of microglia during CNS development. We added the sentences in the discussion section with reference (Hanisch UK, Kettenmann H. Microglia: active sensor and versatile effector cells in the normal and pathologic brain. Nat Neurosci. 2007 Nov;10(11):1387-94. Review. PubMed PMID: 17965659) as below. (P19, line2-5)

 “... for fetal development. In addition, there is a possibility that the increase in TSPO expression during the neural developmental stage might be reflective of activated microglia. During neural development, microglia are activated and maintain CNS homeostasis by removing apoptotic cells, myelin debris and other endogenous redundant debris [40].”

Reviewer 2

1. Thank you for your advice. As you pointed out, the classification of microglia is complicated according to various studies. We described only classical classification of microglia in introduction section and did not mention detailed classification such as M (LPS) and M (IL-4 and IL-13) (previously named as M2a) that reported recently (Michell-Robinson MA et al, Brain. 2015 May;138(Pt 5):1138-59). The objective of this study is clarification of transcriptional mechanism of Tspo gene in mice, hence, we did not intend to mention the classification of microglial cell. LPS and IFN-γ were used for induction of activated microglia and Tspo gene expression as pro-inflammatory inducers.

2. Thank you for your advice. The lists of oligonucleotide sequences and antibodies were transferred to S1 and S2 Table.

3. Thank you for your advice. We revised the manuscript and was proofread by the Springer Nature Author Services.

Reviewer 3

1. Thank you for your advice. The title was replaced as follows. “Lipopolysaccharide induces mouse translocator protein (18 kDa) expression via the AP-1 complex in the microglial cell line, BV-2.”

2. Thank you for your comments. Upon LPS or pro-inflammatory stimulation, various pathways were activated and HDAC1 was released from AP-1 binding site, transcriptional activators were then recruited to the Tspo promoter. Of course, since it is thought that not all HDAC1 molecules were released from AP-1 binding site, some HDAC1 molecules remained at the AP-1 binding site. Furthermore, the enrichment of HDAC1 might gradually returns to normal condition in time-dependent manner after once upon release from AP-1 binding site to inhibit activated gene expression.

3. Thank you for your advice. We revised the manuscript and was proofread by the Springer Nature Author Services.

---

## [Decision Letter · Decision Letter 1]

10 Sep 2019

[EXSCINDED]

Lipopolysaccharide induces mouse translocator protein (18 kDa) expression via the AP-1 complex in the microglial cell line, BV-2

PONE-D-19-15932R1

Dear Dr. Nakamura,

We are pleased to inform you that your manuscript has been judged scientifically suitable for publication and will be formally accepted for publication once it complies with all outstanding technical requirements.

With kind regards,

Leo T.O. Lee, Ph.D.

Academic Editor

PLOS ONE

Additional Editor Comments (optional):

Reviewers' comments:

Reviewer's Responses to Questions

**Comments to the Author**

1. If the authors have adequately addressed your comments raised in a previous round of review and you feel that this manuscript is now acceptable for publication, you may indicate that here to bypass the “Comments to the Author” section, enter your conflict of interest statement in the “Confidential to Editor” section, and submit your "Accept" recommendation.

Reviewer #1: All comments have been addressed

Reviewer #2: All comments have been addressed

2. Is the manuscript technically sound, and do the data support the conclusions?

Reviewer #1: Yes

Reviewer #2: Yes

3. Has the statistical analysis been performed appropriately and rigorously? 

Reviewer #1: Yes

Reviewer #2: Yes

4. Have the authors made all data underlying the findings in their manuscript fully available?

Reviewer #1: Yes

Reviewer #2: Yes

5. Is the manuscript presented in an intelligible fashion and written in standard English?

Reviewer #1: Yes

Reviewer #2: Yes

6. Review Comments to the Author

Reviewer #1: The authors have adequately addressed my comments. The language has been improved and the manuscript is now better readable.

Reviewer #2: (No Response)

7. PLOS authors have the option to publish the peer review history of their article (what does this mean?). If published, this will include your full peer review and any attached files.

Reviewer #1: No

Reviewer #2: No

---

## [Editor Report · Acceptance letter]

12 Sep 2019

PONE-D-19-15932R1 

Lipopolysaccharide induces mouse translocator protein (18 kDa) expression via the AP-1 complex in the microglial cell line, BV-2 

Dear Dr. Nakamura:

I am pleased to inform you that your manuscript has been deemed suitable for publication in PLOS ONE. Congratulations! Your manuscript is now with our production department. 

With kind regards,

on behalf of

Dr. Leo T.O. Lee 

Academic Editor

PLOS ONE